# Syndecans and Pancreatic Ductal Adenocarcinoma

**DOI:** 10.3390/biom11030349

**Published:** 2021-02-25

**Authors:** Nausika Betriu, Juan Bertran-Mas, Anna Andreeva, Carlos E. Semino

**Affiliations:** Tissue Engineering Research Laboratory, Department of Bioengineering, IQS-School of Engineering, Ramon Llull University, 08017 Barcelona, Spain; nausikabetriur@iqs.url.edu (N.B.); juanbertranm@iqs.url.edu (J.B.-M.); annaandreeva@iqs.url.edu (A.A.)

**Keywords:** pancreatic ductal adenocarcinoma, syndecans, proteoglycans, tumor progression, angiogenesis

## Abstract

Pancreatic Ductal Adenocarcinoma (PDAC) is a fatal disease with poor prognosis because patients rarely express symptoms in initial stages, which prevents early detection and diagnosis. Syndecans, a subfamily of proteoglycans, are involved in many physiological processes including cell proliferation, adhesion, and migration. Syndecans are physiologically found in many cell types and their interactions with other macromolecules enhance many pathways. In particular, extracellular matrix components, growth factors, and integrins collect the majority of syndecans associations acting as biochemical, physical, and mechanical transducers. Syndecans are transmembrane glycoproteins, but occasionally their extracellular domain can be released from the cell surface by the action of matrix metalloproteinases, converting them into soluble molecules that are capable of binding distant molecules such as extracellular matrix (ECM) components, growth factor receptors, and integrins from other cells. In this review, we explore the role of syndecans in tumorigenesis as well as their potential as therapeutic targets. Finally, this work reviews the contribution of syndecan-1 and syndecan-2 in PDAC progression and illustrates its potential to be targeted in future treatments for this devastating disease.

## 1. Introduction

The pancreas is an organ that functions as part of the gastrointestinal system. Even though it is primarily an exocrine gland, it also has an endocrine function. The endocrine pancreas is constituted of pancreatic islets, which produce and secrete insulin and glucagon hormones (among others), to regulate blood glucose levels and glucose intake by cells. The exocrine pancreas instead is constituted by the pancreatic duct and acinar cells. It is in charge of producing enzymes such as proteases, lipases, and amylases that are released into the duodenum to support nutrient digestion. Pancreatic dysfunction can lead to digestion problems as well as dysregulation of blood glucose homeostasis due to different diseases including diabetes mellitus, chronic and acute pancreatitis, and hereditary pancreatitis. Moreover, the most relevant for patient’s survival is, without doubt, pancreatic cancer in the form of pancreatic ductal adenocarcinoma (PDAC). PDAC accounts for more than 90% of all pancreatic malignancies and is usually diagnosed at very advanced stages due to the lack of efficient screening tests present for early detection as well as due to its asymptomatic nature at early stages. Moreover, some of its symptoms, like abdominal pain, jaundice, and dark urine are not specific to this disease, which makes it even more difficult to diagnose [1]. The cause of pancreatic cancer remains unknown, but a few authors have recently expressed that some of the risk factors of developing PDAC are cigarette smoking, a diet based on a high intake of fat and meat, diabetes, alcohol abuse, and family history [1,2]. There are three precursors from which PDAC originates [3], namely intraductal papillary mucinous neoplasm (IPMN), mucinous cystic neoplasm (MCN) and pancreatic intraepithelial neoplasm (PanIN), the last one being the most common precursor lesion of pancreatic cancer. PanINs are non-invasive epithelial neoplasms usually located at the head of the pancreas. PanINs can be divided into PanIN-1, PanIN-2, and PanIN-3, depending on its stage. These neoplasms are considered to be the first steps before PDAC development and each one is associated with specific mutations. For example, PanIN-1 is characterized by alterations in Epithelial Growth Factor Receptor (EGFR) signaling and KRAS mutations, while PanIN-2 and -3 are characterized by the inactivation of tumor suppressor genes like CDKN2A, SMAD4, and TP53 [4].

As mentioned above, KRAS mutation is one of the first events in PanIN progression into PDAC. Constitutive activation of mutant KRas promotes plasticity of neoplastic cells and tumor development. Moreover, KRas activates signaling events such as MAPK and PI3K/Akt pathways, which regulate genes involved in cell proliferation, migration, survival/apoptosis, and metastasis [4]. These signaling pathways are normally activated through different cellular receptors such as tyrosine kinase receptors (TKRs), a receptor family which participates in a wide range of cellular processes, like cell proliferation, growth, and invasion [5]. Most of these biochemical pathways, such as cell proliferation, survival and metastasis, are possible because of the extracellular matrix (ECM). The ECM is a three-dimensional network of fibrous proteins, glycoproteins, proteoglycans, and polysaccharides with different biochemical and physical properties synthesized and secreted by stromal cells, mainly fibroblasts. Furthermore, the ECM provides structural and biochemical support for organs and tissues, epithelial cell layers as basal membrane, and individual cells as a substrate for migration [6,7]. The ECM, as well as cell–cell contact and diverse signaling molecules, serves as an information exchanger between cells forming a tissue. These interactions provide the necessary information to preserve cellular differentiation and thus create complex tissue structures. During the early stages of tumorigenesis, the ECM suffers a remodeling that supports tumor initiation and proliferation. This remodeling is caused mainly by enzymes called matrix metalloproteinases (MMP), overexpressed in most human cancers, which hydrolyze the ECM proteins [8].

Pancreatic cancer is characterized by a desmoplastic reaction, in which the deposition of abundant amounts of ECM by stromal pancreatic stellate cells (PSCs) exert biochemical and mechanical effects on PDAC cells [9]. The degree of stromal reaction predicts an aggressive phenotype, and it has been related to chemotherapeutic resistance. In fact, the accumulation of ECM components in PDAC is so high that the stroma accounts for more than 90% of the total tumor mass [10]. For this reason, the pancreatic cancer stroma has attracted the attention of many researchers in terms of targeting the ECM for the development of new PDAC treatments. However, special care should be taken to avoid the depletion of the entire ECM, as this could have dramatic consequences [10]. The degradation of the ECM components and the basal membrane implies the destruction of a physical barrier, which would allow stromal invasion by cancer cells as well as the migration of endothelial cells into the matrix, resulting in neovascularization. Moreover, during the ECM degradation, different growth factors may be released, a fact that would support the proliferation of cancer cells and tumor growth.

Another important component of the ECM which also plays a significant role in cancer cells are the proteoglycans (PGs). PGs are not only restricted to extracellular locations but are also present in cell membranes, acting as receptors to transduce extracellular signals. They are involved in functions like tissue repair and the development and maintenance of homeostatic intracellular processes. The PGs are constituted by core proteins where glycosaminoglycans (GAGs) are covalently attached. There are different types of GAGs but the most important ones are heparan sulfate (HS) and hyaluronan or hyaluronic acid (HA).

Heparan sulfate may be the most complex GAG identified until now, because of modifications and sulfations. As mentioned above, HS can be attached to a proteoglycan, forming heparan sulfate proteoglycans (HSPGs) and both are crucial for cancer initiation and progression. Importantly, HSPGs are able to modulate the interaction between growth factors and TKRs as well as the intracellular transport of extracellular vesicles carrying proteins and nucleic acids implicated in the development of cancer [11,12]. Different studies sustain the importance of these HSPGs in the development and physiology of the organisms, affecting metabolism, transport, and information transfer [13]. The major HSPGs present on the cell surface are syndecans and glypicans. Syndecans are a group of four members classified as “full-time” HSPGs because their function is restricted to the HS chains attached to the PG core protein [14]. The aim of this work is to review the interaction of syndecans with cellular receptors, their role in PDAC progression, as well as their inhibition as a potential PDAC treatment. Also, we will examine the role of syndecan-2 in promoting angiogenesis and its shedding as a key factor for angiogenesis inhibition.

## 2. Syndecan Structure and Interactions

Syndecans are type I glycoproteins encoded by the genes SDC1, SDC2, SDC3, and SDC4, which are expressed throughout the body, participating in different functions and pathways (Table 1) [15]. Syndecan structure can be divided into three differentiated domains, the ectodomain (the N-terminal polypeptide where GAGs are attached), a single transmembrane domain, and the C-terminal cytoplasmic domain (Figure 1a).

The ectodomain is important for cell–cell and cell–matrix interactions via attached GAGs, and introduces variances between syndecan types, unlike the other two domains that are highly conserved. In particular, heparan sulfate (HS) and chondroitin sulfate (CS) are the GAGs present in syndecan-1 and syndecan-3, while for syndecan-2 and syndecan-4, there are exclusively HS chains [42]. As GAGs are able to bind ECM molecules, syndecans, which interact with the cytoskeleton through its cytoplasmatic domain, may promote mechanical cellular responses [43]. Moreover, syndecans cooperate with specific integrins through the core protein to mediate cell adhesion [44,45]. For example, syndecan-1 cooperates with several integrins (αvβ3, αvβ5, α2β1, α3β1, and α6β4) [46,47], while syndecan-2 interacts with β1 integrins [48].

The transmembrane domain is necessary for the dimerization of core proteins into homodimers, which are essential for activating cascade signals [49]. The cytoplasmic domain is directly linked to the cytoskeleton, interacting with several kinases, and promoting different cellular functions [50]. It is divided into three subdomains (Figure 2b), two constant regions (C1 and C2) separated by a variable region (V), which is unique for every syndecan. The C1 region, the membrane-proximal domain, is associated with cytoskeletal interactions and endocytosis [51]. For example, in syndecan-2, the C1 region interacts with ezrin, an actin-associated cytoskeletal protein. Since C1 is a conserved region, the possibility exists that other syndecans could also be able to generate this interaction [52]. In syndecan-3, C1 binds to Src kinase and one of its substrates, cortactin, initiating neurite outgrowth [53]. On the other hand, the C2 region, the membrane-distal domain, interacts with PDZ binding proteins (synbindin, synectin, and syntenin) [50]. In particular, the interaction between syndecans and their adaptor protein syntenin is crucial for the biogenesis and loading of exosomes, a type of secreted vesicle involved in physiopathological processes such as cardiovascular diseases, neurodegeneration, and tumor progression. Exosomes secreted by cancer cells can contribute to tumor progression by fostering angiogenesis and the migration of tumor cells [54].

The V region is where most intracellular interactions between syndecans and other molecules take place. In syndecan-4, this region binds to phosphatidylinositol (4,5)-bisphosphate (PIP_2_) and its binding promotes the activation of PKCα [55,56]. Moreover, recent studies demonstrated that the V region motif (KKXXXKK) acts as a scaffold for PKCα, regulating its localization, activity, and stability [40,57]. This binding also triggers the activation of RhoA and RhoGTPases, and through syndecan-4 cooperation, it promotes focal adhesion formation [58]. The syndecan-4-PKCα interaction might be regulated by the phosphorylation of Ser183 residue [59]. This phosphorylation decreases the affinity of syndecan-4 to PIP_2_ and consequently to PKCα, which in turn decreases and prevents cell migration. However, this phosphorylation promotes an alternative binding of α-actinin to the V region [41,60] which creates a direct linkage between syndecan-4 and the actin cytoskeleton [38]. It has been demonstrated that interactions between the V region and α-actinin and/or PKCα are necessary for signal mechanotransduction and mechanical adaptation to force. Deletion of α-actinin and PKCα binding sites has been associated with defects in cytoskeletal organization, stress fiber assembly, and the inhibition of myofibroblast differentiation [38].

It is well known that syndecans interact with other membrane receptors, acting as co-receptors. They are able to associate with integrins, growth factor receptors (GFRs), as well as adhesion, invasion, angiogenic and migration promoters, and extracellular matrix glycoproteins and collagens [61]. An important aspect of this co-receptor behavior is the interaction with growth factors (GFs) and their respective GFRs. For example, Vascular Endothelial Growth Factor (VEGF) plays a key role in blood vessel formation and maintenance during development. Syndecans are believed to act as Vascular Endothelial Growth Factor Receptor (VEGFR) co-receptors by binding to its ligand and increasing its local concentration in the cell plasmatic membrane, facilitating their binding to VEGFR [62]. Indeed, in the absence of the HS chains that are present in syndecans, the VEGF would not be able to find some of its membrane receptors. Syndecans also promote ligand-receptor binding in the case of Fibroblast Growth Factor (FGF) and its receptor, where the O-sulfation pattern of the HS chains seems to be crucial for its binding capacity to FGF [63,64].

The Epithelial Growth Factor Receptor (EGFR) also interacts with the ectodomain of syndecans. In particular, Chronopoulos et al. [38] demonstrated that when inhibiting the EGFR with Gefitinib, pancreatic stellate cells (PSC) were unable to react to external tension applied to syndecan-4, preventing adaptive cell stiffening and also force-induced PI3K activation. They concluded that syndecan-4, in cooperation with EGFR and β-1 integrin, tunes cell mechanics in response to localized tension via kindlin-2 and RhoA in a PI3K-dependent manner (Figure 2a). In another work, Wang et al. [65] described, using HaCaT and MCF10A epithelial cell lines, that syndecan-1 and syndecan-4 directly interact with HER-2 and EGFR, respectively, and also capture α3β1 integrin. This ternary complex interacts with α6β4 integrin via its cytoplasmatic link with the syndecan, leading to activation and downstream signaling that promote motility and survival (Figure 2b) [65]. Other described interactions include the association between syndecan-1, the insulin-like growth factor-1 receptor (IGF1-R) and αvβ3 or αvβ5 in both human mammary carcinoma cells and endothelial cells undergoing angiogenesis. This mechanism requires syndecan-1 clustering or engaging with matrix ligands. When captured by syndecan-1, IGF1-R suffers autophosphorylation and activation. This initiates an inside-out signaling that activates talin, which in turn promotes integrin activation. This pathway can be competitively blocked by the Synstatin (SSTN_92–119_), a short peptide that displaces IGF1-R and integrins from syndecan-1, preventing its activation [66,67,68] (Figure 2c).

Other examples include the cooperation between neuronal Thy-1, syndecan-4, and ανβ3 integrin in astrocyte cells to activate PKCα and in consequence, the RhoA pathway. This response was inhibited when SDC4 expression was silenced or a syndecan-4 mutant lacking the intracellular domain was overexpressed [69]. Syndecan-4 and α5β1 integrin also cooperates to activate PKCα in melanoma cells [70]. Moreover, the Thy-1-integrin linkage is relevant in melanoma invasion, myocyte transmigration through endothelial cells, and host defense mechanisms. In fact, the triple cooperation between Thy-1, syndecan-4, and α5β1 integrin is responsible for mediating the contractility response to mechanosignals in melanoma cells [71].

Finally, the interaction of syndecans with integrins can also be in an indirect way via intermediate receptors. For example, syndecan-2 has been described to interact with the protein tyrosine phosphatase receptor CD148 to promote Src and PI3K signaling, which in turn regulate β1 integrin-mediated adhesion processes, like angiogenesis and inflammation [48].

## 3. Syndecan Shedding

Syndecans are usually cleaved near the membrane by a process called shedding. The extracellular domain of the syndecans is released to the extracellular space and is converted into a soluble effector that can bind to growth factors, extracellular ligands, membrane receptors, etc. During this event, the GAGs are even more important because they interact with a great percentage of molecules in this shed-syndecan form. This highly regulated process is carried out by matrix metalloproteinases (MMPs), also known as sheddases of syndecans, which usually cleave before a hydrophobic residue. In humans, 23 different MMPs are known to exist, 14 of them expressed in vasculature. MMPs are produced by multiple tissues and cells: connective tissue, pro-inflammatory, and uteroplacental cells, fibroblasts, osteoblasts, endothelial cells, vascular smooth muscle cells, macrophages, neutrophils, lymphocytes, and cytotrophoblasts. MMPs are regulated at many levels, including at the mRNA and protein level (this last one by activation of the proenzyme to its active form), but also by growth factors. For example, VEGF-A downregulation promotes a decrease in MMP-2 expression [72], while the presence of Epithelial Growth Factor (EGF) upregulates MMP-1 and MMP-9 transcripts [73]. However, the most important regulators of MMPs are the tissue inhibitors of matrix metalloproteinases (TIMPs), which coexist in balance with MMPs. When MMPs are upregulated or TIMPs are downregulated (compared to homeostatic values), an imbalance that can lead to various pathological conditions such as heart failure, osteoarthritis, and cancer is produced [74]. MMPs activity is difficult to detect, except for during tissue repair processes such as wound healing and menstruation [75]. Some of the MMPs are excreted as proMMPs and then activated on the cell surface by various factors such as heat, low pH, chaotropic agents, and thiol-modifying agents, like N-ethylamine [74]. MMP expression or activity is influenced by hormones, GFs, and cytokines [76]. For example, in the case of menstruation, as a clear example of MMP action, the ovarian sex hormones regulate the expression of these MMPs during endometrium tissue remodeling. MMPs are important in many biological processes, such as cell proliferation, migration and invasion, ECM remodeling, and vascularization. These processes are constantly occurring in the whole body, but if not controlled, they can lead to different diseases such as cancer. MMPs participate in these pathological processes by cleaving GFs and PGs and by playing a role in ECM degradation.

Syndecans are a target for shedding by MMPs and this process has multiple effects on cell signaling (Figure 3). As extracellular soluble molecules, shed syndecans can bind ligands, preventing them from finding their transmembrane receptor. Moreover, as soluble effectors, they can start new signaling pathways through binding to other transmembrane receptors [77]. The syndecan shedding through MMPs is known to be induced by GFs [78], chemokines [79], bacterial virulence factors [80], trypsin [81], insulin (especially for patients with diabetes mellitus) [82], heparinase [83], and cell stress [74]. Intracellular mechanisms can also lead to syndecan shedding [84], as the tyrosine residues of the cytoplasmic domain are an important site of phosphorylation to induce this process [78,84]. But why and when does this shedding happen? It is known that syndecan shedding happens mainly in three main processes: wound healing, cancer, and bacterial pathogenesis.

### 3.1. Syndecan Shedding during Wound Healing

Wound healing is a biological process divided into three steps, inflammation, proliferation, and regeneration. Syndecans, as a major source of GAGs, control a large number of cytokines through GAG binding. For example, they are involved in leukocyte recruitment [85] and during inflammation, syndecan expression is upregulated [86]. In a recent study, the authors reported an upregulation of syndecan-2 in response to acute-induced inflammation in mice colons. This study also reports high expression levels of syndecan-2 in other inflammation-associated diseases, like colitis and sepsis, as this high-level expression is a good marker for acute inflammation and acute inflammation-associated diseases [87]. Another study using human umbilical vascular endothelial cells (HUVECs) demonstrated that syndecan-4 expression increased after inducing inflammation using lipopolysaccharides (LPS) and IL-β1 (which together mimic a bacterial infection and a general inflammatory condition). The authors also generated SDC4 knock-down endothelial cells, in which the absence of syndecan-4 in the membrane prevented the MMP from cleaving its ectodomain, delaying wound healing and tube formation [88]. This was described before using a mice model in which SDC4 knock-out (KO) induced wound healing deficiencies and impaired angiogenesis (Table 2) [89]. In another study, Chen et al. [90] reported that in MMP-7 null mice, the epithelial cells were unable to repair the wound after an injury, demonstrating the importance of syndecan-1 shedding by MMP-7 in re-epithelization after injury [90].

### 3.2. Syndecan Shedding during Tumor Development

Tumor development implies the participation of oncogenic genes, but also its growth, invasion, and spreading through metastasis is the result of the cooperation between GFs, integrins, TKRs, and syndecans. Syndecans can promote cell proliferation, survival, and invasion through different signaling pathways, for instance, by the activation of the KRas/MAPK pathway by syndecan-2 or the Wnt pathway by syndecan-1. These examples represent syndecans acting as membrane receptors, but as mentioned above, this type of proteoglycans can also act as soluble molecules when shed from the surface of the cell membrane. Syndecan shedding has implications in tumor progression, especially in metastasis. Soluble syndecans have attracted the attention of many researchers with the aim of establishing correlations between them and tumor progression. For example, in heparinase-induced shedding by myeloma cells, the soluble syndecan-1 promotes endothelial cell invasion and angiogenesis. This is achieved by the binding of syndecan-1 HS chains to VEGF [91]. A recent study involving breast cancer patients demonstrates that syndecan-1 levels in its soluble form increased compared to healthy patients [19]. They also report that these high levels of soluble syndecan-1 correlate with tumor size. This correlation has unknown implications, but it could serve as a serum biomarker for breast cancer. Another example is the work done by Szarvas et al. [20], in which they analyzed the levels of soluble syndecan-1 in the serum of prostate cancer patients and found out it was higher compared to the control group [20]. Although they could not explain the reason for these increased levels, they maintain that the syndecan-1 ectodomain present in the serum could be used as a biomarker to improve the prognosis of prostate cancer. Indeed, syndecan-1 can be used as a biomarker for a lot of tumors, such as lung cancer, hepatocellular carcinoma, and bladder cancer, among others [92]. 

### 3.3. Syndecan Shedding during Bacterial Pathogenesis

Some bacterial pathogens take advantage of syndecan shedding in order to inhibit the immune response of the host and enhance their own pathogenicity. For example, *Staphylococcus aureus*, a Gram-positive bacterium responsible for causing different diseases like pneumonia, toxic shock syndrome, osteomyelitis, and sepsis, generates syndecan-1 shedding through the α- and β-toxin, the latter one also being able to induce syndecan-4 shedding. Even though these toxins do not promote syndecan shedding directly, they stimulate signaling pathways that induce it, such as the activation of tyrosine kinase receptors [93]. Another example would be *Pseudomonas aeruginosa*, an opportunistic Gram-negative bacterium that induces syndecan-1 shedding through the protease LasA [94]. Therefore, considering the role of syndecan shedding in promoting bacterial pathogenesis, the authors suggest the use of shedding inhibitors to treat bacterial infections.

## 4. Syndecans in Cancer

In addition to playing many roles in development and signaling under physiological conditions, syndecans are also important in the progression of some malignancies. Their expression levels increase during cancer progression and therefore they can be considered good candidates for possible prognostic markers. For example, syndecan-1, also known as CD138, has been described to be present in multiple myeloma [95], breast [18,19], colorectal [21], and pancreatic carcinomas [22,23], among others. In highly metastatic breast cancer, syndecan-1 levels are higher than in those with low metastatic character, suggesting that syndecan-1 could be a remarkable biomarker for metastatic breast cancers. In particular, syndecan-1 promotes tumorigenesis via the Wnt pathway. Generation of SDC1-null transgenic mice, crossed with Wnt1-expressing mice in the mammary gland, showed that Wnt1-induced hyperplasia was reduced by 70% and that the Wnt pathway was inhibited (Table 2) [18]. Syndecan-1 also promotes cell proliferation, adhesion, and angiogenesis in mouse embryonic fibroblasts (MEFs). When co-cultured with highly invasive carcinoma cells, those MEFs that were SDC1^+/+^ enhanced the growth of carcinoma cells by 40% compared with SDC1^−/−^ MEFs [96]. Syndecan-1 has also been used as a target for multiple myeloma treatment. In particular, Indatuximab ravtansine (BT062), an antibody-drug conjugate that binds to syndecan-1, has recently been successfully used in clinical trials in patients with relapsed multiple myeloma, stabilizing or improving disease in almost 80% of patients [97].

Syndecan-2 is associated with breast cancer’s metastatic ability (angiogenesis and neovascularization), morphology, and invasion index, in part by regulating RhoGTPases [98]. It is a survival predictor for head and neck cancer [99] and is also associated with colorectal cancer, in which it has implications in the migratory behavior of highly metastatic tumor cells [28].

Syndecan-3 enhances epithelial-to-mesenchymal transition (EMT) in metastatic prostate cancer, suggesting that its attenuation, and consequently its signaling pathways, could lead to a better therapeutic outcome in prostate cancer [100]. As other syndecans, syndecan-3 has a role in angiogenesis, with a high expression in tumoral stromal vessels [101].

Finally, syndecan-4 is known to be expressed in breast cancer, regulating cell adhesion and spreading and also interacting with GFRs. For example, the estradiol–estrogen receptor complex initiates the growth and progression of hormone-dependent breast cancer, and it seems that syndecan-4 participates in this pathway as a mediator factor, being activated by the Insulin-like Growth Factor Receptor (IGFR) [102]. In addition, the expression levels of SDC4 in human breast cancer links with the FGF2R complex formation, indicating that syndecan-4 regulates this FGF2R complex formation in human breast cancer [103].

### 4.1. Syndecans in PDAC

#### 4.1.1. Syndecan-1

Pancreatic ductal adenocarcinoma is one of the most lethal human pathologies, mainly because of its late detection and metastatic capacity [3,104]. It is the only gastrointestinal pathology showing syndecan-1 upregulation [22]. This overexpression correlates with cell proliferation, differentiation, and invasion for the development of PDAC. The stroma of PDAC has increased levels of syndecan-1 compared to a normal stroma. Indeed, the shift of syndecan-1 expression from the epithelium to the stroma is a poor prognostic factor in PDAC [105]. It has also been suggested that syndecan-1 performs different functions depending on its location: epithelial syndecan-1 promotes an epithelial morphology while stromal syndecan promotes tumorigenesis [105]. Heparanase (HPA) is an endoglycosidase able to specifically degrade the HS chains of syndecan-1. It promotes tumor progression and metastasis by enhancing the synthesis and shedding of syndecan-1 [83]. HPA is overexpressed in pancreatic cancer [106] and its expression has been correlated with cancer cell invasion and lymph node metastasis in PDAC patients [107,108]. Moreover, HPA modulates the response of pancreatic cancer to radiotherapy. In particular, ionizing radiation (IR) upregulates HPA expression, promoting the invasive ability of pancreatic cancer cells in vitro and in orthotopic tumors in vivo. This could be one explanation not only for tumor resistance to radiotherapy but also for its effect in enhancing tumor dissemination. Combined treatment with an HPA inhibitor and IR attenuated spreading in orthotopic pancreatic tumors in mice [109]. The HPA/syndecan-1 axis promotes the upregulation of FGF2, which in turn activates the PI3K/Akt pathway and EMT in cultured pancreatic cancer cell lines [110]. At the same time, FGF2 also promotes syndecan-1 shedding via MMP7 in Panc-1 cells [111]. Moreover, syndecan-1 shedding in this cell line is enhanced by treatment with the chemotherapeutic drugs bortezomib and doxorubicin [112]. Increased levels of shed syndecan-1 have been reported in some cancers such as breast [19] and prostate [20] and have been correlated with poor prognosis in patients with lung cancer [113] and myeloma [91]. To our knowledge, there are no in vivo studies reporting the shedding of syndecan-1 in pancreatic cancer. However, the fact that HPA, FGF2, and MMP7 are all upregulated in PDAC tissue samples [110,114,115] makes it reasonable to assume that syndecan-1 shedding could also be happening during pancreatic cancer progression. This is further supported by the fact that in pancreatic cancer, epithelial syndecan-1 is produced by the epithelial cancer cells [22], but the origin of the stromal syndecan is unknown. This stromal syndecan-1 could be produced by mesenchymal cells or could be shed from epithelial cells into the tumor stroma. It is possible that the release of syndecan-1 to the stromal compartment contributes to the pancreatic malignant phenotype by binding to growth factors that support cell proliferation, as happens in breast cancer [116,117].

The KRAS mutation is the most frequent mutation in PDAC and is believed to be an initiating step for pancreatic carcinogenesis. It seems that 95% of late-stage pancreatic cancers present with a mutated and highly overexpressed KRAS. Interestingly, syndecan-1 has been described to cooperate with KRas to induce the malignant phenotype. In particular, a recent study demonstrates that syndecan-1 expression serves as a KRas effector, inducing macropinocytosis in PDAC [23]. Macropinocytosis is a type of endocytosis, which involves the non-specific intake of extracellular material, like soluble molecules, nutrients, and antigens. The study demonstrated that in low-glutamine medium, upregulated KRas cells decreased proliferative capacity. In the presence of albumin as a substitute of glutamine, cell proliferation was rescued as albumin was incorporated into the cell by macropinocytosis. SDC1 knock-out cells reduced albumin intake capacity and consequently, reduced cell proliferation in low-glutamine conditions [23]. Thus, this new study reveals that syndecan-1 plays a crucial role in macropinocytosis in KRAS-driven pancreatic cancer. In another study, the authors investigated whether there was an association between SDC1 and KRAS expression and patient survival. They found that patients carrying KRAS somatic mutations had a higher SDC1 mRNA expression than those without mutations, and that this gene signature elevated mortality [118]. Both studies suggest that targeting KRAS and SDC1 in combination could improve patient outcomes [23,118].

#### 4.1.2. Syndecan-2

Syndecan-2 also plays a significant role in pancreatic cancer, working as an invasive-associated gene that, as well as syndecan-1, cooperates with KRas to induce the invasive phenotype [27]. Oliveira et al. [27] reported high syndecan-2 expression levels in various pancreatic cancer cell lines, like T3M4, Panc-1, and SU8686. When SDC2 was silenced by iRNA, migrating and invading cells were reduced significantly, although cell growth was not. In a similar way, some of the KRas/MAPK signaling pathway components, like phosphorylated Src and phosphorylated ERK, were also reduced when reducing SDC2 expression levels. Therefore, they demonstrated that syndecan-2 is an important mediator in PDAC and that it cooperates with KRas to increase malignancy and perineural invasion. The upregulation of syndecan-2 indirectly interferes with the KRas/MAPK signaling pathway enhancing the mutated KRAS gene upregulation and in consequence, the Ras protein GTP-phosphorylation. In particular, p120-GAP and RACK1 proteins compete for the binding to the syndecan-2 cytoplasmatic domain [119]. RACK1 plays an important role regulating the phosphorylation of Src and preventing it when Src is associated to syndecan-2. However, when p120-GAP, instead of RACK1, binds to syndecan-2, Src is phosphorylated using this binding as a switch signal [119]. When phosphorylated, Src interacts with several receptors, G-proteins, signal transducers, and transcription molecules, resulting in biological functions involving cell proliferation, growth, and differentiation [120]. The early steps for p120-GAP binding to syndecan-2 are related to the KRAS mutation. It has been demonstrated that in pancreatic cancer cell lines with wild-type KRAS, RACK1 and syndecan-2 are interacting. Alternatively, p120-GAP binds to syndecan-2 in those cell lines with mutated KRAS [27] (Figure 4). This study shows the importance of syndecan-2 in pancreatic ductal adenocarcinoma and its cooperation with oncogenic KRAS gene to aggravate this malignant phenotype. Even though syndecan-1 and -2 participate in the regulation of different processes in PDAC, both enhance KRas signaling. This fact opens a new window in PDAC treatment, as syndecan targeting could downregulate the KRas/MAPK pathway.

#### 4.1.3. Syndecan-3 and -4

Syndecan-3 is also increased in PDAC and its expression has been positively correlated with tumor size in an orthotopic mice model [121]. It is associated with Midkine (MK), a type of neurotrophic factor that triggers different responses, like neurite outgrowth, neuronal survival, carcinogenesis, and tumor progression. It has been shown that the interaction between MK and syndecan-3 generates perineural invasion and poor prognosis [122].

Less is known about the presence and role of syndecan-4 in the pancreas. It has been detected in pancreatic islet β-cells of mice, rats, and humans, and also in the pancreatic β-cell line MIN6 [123,124]. Importantly, syndecan-4 expression was found to be negative in other pancreatic islet cells as well as in exocrine cells, suggesting its specific role in the regulation of insulin secretion in β-cells. Moreover, SDC4 mRNA expression in β-cells is transiently upregulated by IL-1β via the Src-STAT3 pathway [123]. The presence and/or dysregulation of syndecan-4 has not been specifically described in PDAC and it does not seem to be relevant. However, even if not described in pancreatic duct epithelial cells, it is present in activated cultured pancreatic stellate cells (PSCs) [38]. PSCs are responsible for producing the desmoplastic stroma in which cancer cells are embedded, and therefore, for promoting PDAC progression [125]. Syndecan-4 has been involved in focal adhesions (FAs) formation through the binding and activation of PKCα [126]. The absence of syndecan-4 not only generates smaller FAs, but also shows an impaired actin-cytoskeleton and defective smooth muscle actin incorporation [127]. Chronopoulos et al. [38] demonstrated that applying an apical force to induce a syndecan-4 response in PSCs generates an increase in talin-1 and kindlin-2 basal accumulation. These are both focal adhesion proteins that bind the integrin cytoplasmic domain, recruit cytoskeletal, and signaling proteins involved in mechanotransduction, and enhance the integrin activation for ECM binding [128]. However, this recruitment requires PI3K action. Localized force on syndecan-4 triggers PI3K activation, which enriches talin-1 and kindlin-2 concentration inducing a cell-global response, even at distal sites from the force application point. A direct link between the presence of syndecan-4 in PSCs and pancreatic cancer progression has not been described yet. However, considering the function of syndecan-4 in regulating focal adhesion formation [39] and cytoskeleton organization [40,41], and since pancreatic cancer tissue can be several folds stiffer than its healthy counterpart [129,130,131], it would be interesting to study the role of syndecan-4 in modulating the mechanical response to the increased tissue stiffness.

## 5. Syndecan-2 in Angiogenesis

Angiogenesis is the process through which new blood vessels are formed from preexisting ones. It mainly occurs during embryonic development, wound healing processes, and organism growth, in which it is highly regulated. However, angiogenesis is present in many pathologies such as rheumatoid arthritis, diabetes, tumor growth, and metastasis, acting as a non-regulated process [132]. Angiogenesis depends on the final balance between pro- and antiangiogenic molecules. In normal tissues, antiangiogenic factors predominate over the proangiogenic factors. In malignant cells, an event called the “angiogenic switch” breaks the normal equilibrium between pro- and antiangiogenic factors, boosting the angiogenic process [133]. Blood vessels have a fundamental role in tumor metastasis to other organs [134], and therefore, the inhibition of angiogenesis is a promising anti-cancer strategy.

Syndecan-2 is a key angiogenic element [25]. Its downregulation in human endothelial cells prevents angiogenesis, reducing cell adhesion and spreading [26]. In fact, syndecan-2 expression represents 70% over the total HSPGs content in human microvascular endothelial cells (HMVECs) [26]. Syndecan-2 expression is usually regulated by growth factors [135]. It has been reported that treatment of microvascular endothelial cells with FGF, VEGF, and PMA increases syndecan-2 expression, and that FGF was the growth factor that enhanced this expression the most, probably due to its co-receptor behavior in some signaling pathways [136]. Moreover, SDC2 gene silencing resulted in reduced cell proliferation and spreading, while enhancing cell migration. Angiogenesis is a multi-step process which includes endothelial cell signaling for the activation of cytoskeletal rearrangements and gene transcription. Moreover, syndecan-2 seems to be involved in this whole process by activating cell proliferation and adhesion, which are necessary for the generation of new blood vessels. Further studies demonstrated that syndecan-2 downregulation generates a decrease and reorganization of focal adhesions. This happened while actin was increasing its assembly, impairing a migratory behavior [137].

One of the principal roles of syndecan-2 in endothelial cells is the interaction with growth factors (VEGFA and FGF) and promotion of signaling pathways via their receptors, VEGFR and FGFR [26]. VEGFR2 is the main signal receptor for VEGFA_165_, a well-known circulating heparin-binding VEGFA isoform. Syndecans act as co-receptors for growth factors binding, increasing the membrane local concentration and enhancing the binding to VEGFR2 [62,138]. Surprisingly, this Syndecan-2-VEGFA_165_-VEGFR2 interaction complex had never been studied until Corti et al. [139] generated a global and endothelial-specific SDC2 knock-out mouse, where this trimolecular complex association was studied [139]. While SDC2 knock-down in zebrafish caused severe problems during embryonic development [25], SDC2^−/−^ mice were born alive. Deep analysis of mice development showed a reduction in retinal vessel outgrowth and decreased vascular branching. Moreover, these mice showed problems with wound healing (in which angiogenesis is a key process) (Table 2). As syndecan-4 structurally and evolutionarily is almost identical to syndecan-2 [140], the authors performed the same experiment for SDC4^−/−^ mice lines. Whereas SDC2^−/−^ mice showed retinal vascular problems, SDC4^−/−^ did not. To reveal the importance of VEGFA_165_ as the key growth factor interacting with syndecan-2, they performed a cornea assay model in which angiogenic response to growth factor addition was examined. They also studied the function of VEGFR2 and syndecan-2 in endothelial cells (ECs) and human umbilical vein endothelial cells (HUVECs). The results suggested that VEGFA_165_ needs syndecan-2 cooperation to enhance VEGFR2 activation. VEGFA_165_ binds to syndecan-2 ectodomain, which facilitates the posterior binding to VEGFR2, generating the syndecan-2-VEGFA_165_-VEGFR2 trimolecular complex (Figure 5a). Further experiments were focused on elucidating through which syndecan domain this molecular interaction happened. They found that the complex formation was due to the binding of VEGFA_165_ to the HS chains of syndecan-2, in particular the 6-O sulfations of heparan sulfate, which was previously reported [141,142]. Collectively, all these considerations demonstrate that syndecan-2 is a crucial angiogenic element, and therefore, syndecan-2 shedding could act as an angiogenic inhibitor. To demonstrate this, Whiteford et al. [143] generated an expression system in HEK293T cells in which the syndecan-2 ectodomain (S2ED) was constitutively released from the cells, and the anti-angiogenic properties of the shed syndecan-2 in a xenograft tumor model were investigated. Mice injected with empty vector cells (control) showed larger tumors compared to those mice injected with HEK293T cells expressing the released syndecan-2 ectodomain. They also demonstrated the inducing activity of TNF-α to shed syndecan-2 and that the anti-angiogenic activity of syndecan-2 was exclusively on its shed form. Moreover, 3D co-cultures of fibroblasts and HUVECs in the presence of S2ED showed a significant reduction in tubule length and branching compared to controls, demonstrating that shed syndecan-2 inhibited endothelial cells invasion through the direct inhibition of cell migration. In particular, S2ED binds CD148 receptor in endothelial cells and reduces the angiogenic response by inactivating β1-integrin (Figure 5b). It is remarkable how syndecan-2 can induce contrary cell signals. When working as a cell membrane receptor, it promotes angiogenesis acting as a co-receptor promoting the association of some ligands to bind its receptors [139]. However, when shed from the cell surface, the syndecan-2 ectodomain can trigger an anti-angiogenic pathway through the inactivation of β1-integrins driven by the association with CD148 [143].

**Table 2 biomolecules-11-00349-t002:** Mice models to study syndecan function.

Mice KO	Phenotype	Conclusion
SDC1^−/−^	Reduced Wnt1-induced hyperplasia and inhibition of the Wnt pathway	Syndecan-1 promotes tumorigenesis via the Wnt pathway in breast cancer [18]
SDC2^−/−^	Retinal vascular problems Deficient wound healing	Syndecan-2 is a key angiogenic element [139]
SDC4^−/−^	Delayed skin wound healing and angiogenesis after injury	Syndecan-4 plays important roles in wound healing and angiogenesis [89]

## 6. Future Perspectives

Pancreatic Ductal Adenocarcinoma is a fatal malignancy with a high mortality rate and an extremely low five-year survival rate of 9% [144]. The initial diagnosis is usually done when the cancer is really advanced, leading to poor prognosis [1]. For an improvement in early PDAC diagnosis, there is a need to find novel biomarkers and to be able to treat the pancreatic cancer in the early stages. In the late 1980s, glycan carbohydrate antigen (CA 19-9) was found to be present at high concentrations in the serum of pancreatic cancer patients [145]. Nowadays, it is the most clinically used biomarker for PDAC detection [146]. Additional biomarkers such as plasma thrombospondin-2 (THBS2) are being studied. THBS2 is a promising biomarker that enables distinguishing between stages of PDAC carcinogenesis. Moreover, combination of CA 19-9 and THBS2 biomarkers improves the discrimination between PDAC patients from those with chronic pancreatitis [147].

Four genes are identified as the predominant effectors of PDAC: KRAS, CDKN2A, TP53, and SMAD4 [4]. KRAS mutation is one of the first events in PanIN progression into PDAC. Constitutive activation of KRas protein promotes tumor development through the activation of signaling events such as the MAPK and PI3K/Akt pathways [4]. Even though mutant KRAS is not a suitable target for PDAC therapy [148], the inhibition of some of its activators or mediators could serve as treatment. In this way, syndecan-1 and syndecan-2, even at different levels, both participate in KRas signaling. It has been recently described that syndecan-1 is a critical mediator of macropinocytosis in KRAS-driven pancreatic cancer [23]. Pharmacological inhibition of macropinocytosis has not been achieved to date. However, this new finding opens a window and calls for targeting syndecan-1 as a therapeutic treatment to inhibit macropinocytosis in pancreatic cancer cells, and thus preventing them from nourishing.

Because syndecan-1 is overexpressed on the surface of different cancer cells, antibody-drug conjugates are considered as an option to target cells expressing this syndecan. In particular, indatuximab ravtansine (BT062) is a monoclonal antibody linked to the cytotoxic agent DM4 (ravtansine) that targets syndecan-1-expressing cells. The internalization of the conjugate by the target cell and the subsequent DM4 release results in its activation and cytotoxic activity. Moreover, its high specificity results in low systemic toxicity. BT062 has been used in different clinical trials to target multiple myeloma (ClinicalTrials.gov (accessed on 1 March 2021), Identifier: NCT00723359 and NCT01001442) [97], triple negative metastatic breast cancer and metastatic bladder cancer (ClinicalTrialsRegister.eu (accessed on 1 March 2021), Identifier: 2013-003252-20).

Synstatin (SSTN_92-119_) is a short peptide that competitively blocks the triple interaction between syndecan-1, αvβ3 or αvβ5 integrins, and IGF-1R, preventing the activation of these last two receptors [66,67,68]. Synstatin has been demonstrated to be a potent inhibitor of mammary tumor growth and angiogenesis in a xenograft mice model with no evident toxic effects [66]. Thus, it is a promising therapeutic agent for diseases that involve αvβ3 or αvβ5 integrins and IGF1-R. It is possible that this triple interaction between syndecan-1, αvβ3 or αvβ5 integrins, and IGF1-R may also be present in pancreatic cancer, since these receptors are also expressed in this kind of tumor [149,150], and if so, synstatin could also be a possible agent for the treatment of PDAC.

The SDC1 gene has been recently reported to be a target of the microRNA-494 [151]. miR-494 decreased mRNA and protein expression levels of syndecan-1 in the pancreatic cancer cell line SW1990 and inhibited EMT, cell proliferation, and migration in vitro. Moreover, it delayed tumor growth in a xenograft mouse model. This study may provide the basis for the application of miR-494 in pancreatic oncology. However, miR-494 also targets other genes such as FOXM1, SIRT1, and c-Myc [152,153], and therefore, further studies would be needed before it can be considered a therapeutic approach against pancreatic cancer.

Instead of being used as a direct target for inhibition, syndecan-1 could also be useful to target the pancreatic tumor itself. For example, syndecan-1 probes have been used to localize and image orthotopic pancreatic tumors in mice using multispectral optoacoustic tomography (MSOT) [154]. This technique promises to be useful in monitoring tumor development, location, and response to therapy, and syndecan-1 has been demonstrated to be a good target for imaging of pancreatic tumors with minimal probe accumulation in off-target organs [154]. The fact that syndecan-1 accumulates in pancreatic tumors could also be exploited for nanoparticle direction to the pancreas for drug delivery.

Syndecan-2 also has a high potential as a therapeutic target for PDAC treatment. Its expression in this type of cancer has been demonstrated [155] and its roles are related to many tumorigenic aspects. Importantly, syndecan-2 enhances pancreatic cancer cell invasion via the Ras/MAPK pathway [27]. Moreover, its association to the actin cytoskeleton confers the ability to respond to localized mechanical tensions, inducing a global response that regulates cell mechanics through different pathways that promote cell adhesion, migration, and invasion. Syndecan-2 also plays key roles in angiogenesis, promoting the binding of VEGFA_165_ to its receptor VEGFR2 [139], and therefore it may be necessary for tumor vascularization. Syndecan-2 downregulation could contribute to the inhibition of tumor growth, angiogenesis, cell invasion and metastasis. Unfortunately, to date, a way to target syndecan-2 has not been described, so more research in this direction would be valuable.

It is noteworthy that most of the in vitro experiments that aim to study syndecan interactions and signaling are performed in traditional 2D cell cultures, which do not recreate the biological ECM architecture. Given the complex interactions within a tissue, important biological features may be missed if they are only studied in unnatural and constraining 2D cell cultures. It has been extensively reported that cell proliferation, morphology, behavior, and signaling are dramatically modified when culturing cells in 2D versus 3D conditions [156]. Therefore, there is a need for 3D cell models that could bridge the gap between 2D cell culture and animal models. Gagliano et al. [157] studied the effect of culturing HPAF-II, HPAC, and PL45 pancreatic ductal adenocarcinoma cell lines in 3D spheroids. They found that SDC1 expression increased in HPAF-II 3D spheroids while it decreased in HPAC spheroids and remained constant in PL45 spheroids compared to 2D cultures. The expression pattern of HPA was also cell line-dependent, while MMP7 was upregulated in 3D spheroids compared to 2D culture in all the cell lines analyzed [157]. This study highlights the importance of 3D cultures to better mimic the malignant phenotype of pancreatic cancer cells found in vivo. 3D cultures not only provide another dimension but also can contribute to mimicking the tumor microenvironment [158]. For example, in 2D cultures biomolecules diffuse away very quickly while in 3D nanofiber-based scaffolds, growth factors bind to the matrix fibers [159] and are likely to establish a local molecular gradient, which is critical for studying syndecan function as co-receptors in a closer real scenario. Therefore, the development of new 3D cancer models that better mimic not only the in vivo phenotype of pancreatic cancer cells but also its microenvironment could help to identify new signaling pathways involved in pancreatic tumorigenesis.

## Figures and Tables

**Figure 1 biomolecules-11-00349-f001:**
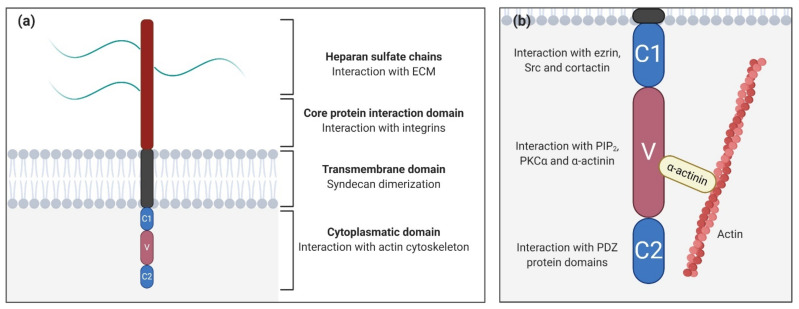
Syndecan structure. (**a**) Schematic representation of syndecan domains and their functions; (**b**) Close-up of the syndecan cytoplasmatic domain showing the constants (C1/C2) and the variable (V) regions and their potential interactions. The interaction between the variable region with α-actinin links syndecans to the cytoskeleton.

**Figure 2 biomolecules-11-00349-f002:**
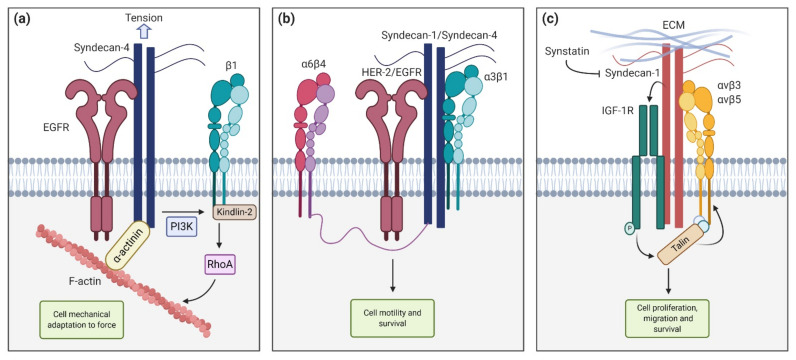
Syndecan functions and interactions. (**a**) Syndecan-4 cooperates with Epithelial Growth Factor Receptor (EGFR) and β1-integrin to tune cell mechanics in response to tension (**b**) Syndecan-1 and syndecan-4 interact with HER-2 and EGFR respectively, and with α6β4 and α3β1 to promote cell motility and survival (**c**) Syndecan-1 interaction with insulin-like growth factor-1 receptor (IGF1-R) activates talin, which in turn activates αvβ3 and αvβ5 integrins, leading to cell proliferation, migration, and survival. These interactions can be blocked with the synstatin peptide.

**Figure 3 biomolecules-11-00349-f003:**
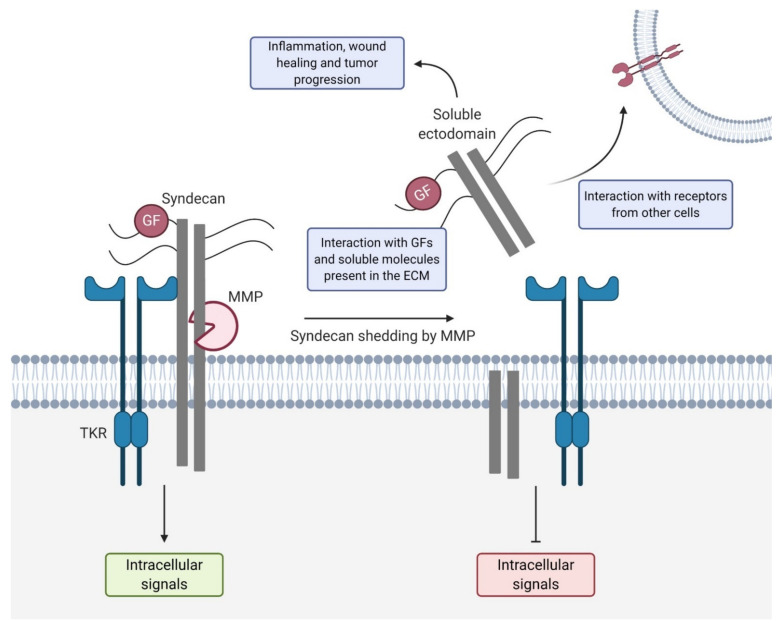
Syndecan shedding. Matrix metalloproteinases (MMPs] cleave syndecans near the transmembrane domain. Syndecan shedding prevents the transduction of intracellular signals that otherwise would be activated by the syndecan ectodomain. The soluble shed syndecan can bind to growth factors, extracellular matrix (ECM) substrates, and also receptors from other cells, which generate new signals or modify others. Soluble syndecans have implications in inflammation, cancer progression, angiogenesis, and wound healing.

**Figure 4 biomolecules-11-00349-f004:**
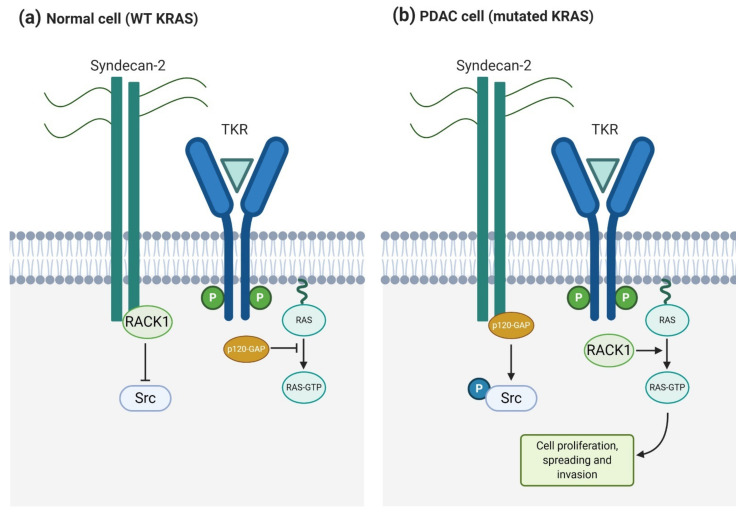
Syndecan-2 and KRas cooperate to induce an invasive phenotype in pancreatic cancer cells. (**a**) In normal cells, RACK1 binding to syndecan-2 prevents Src activation and free p120-GAP inhibits Ras signaling triggered by TKRs (**b**) In pancreatic ductal adenocarcinoma (PDAC) cells, p120-GAP binding to syndecan-2 activates Src and free RACK1 enhances TKR-mediated Ras activation, promoting cell proliferation, spreading, and invasion.

**Figure 5 biomolecules-11-00349-f005:**
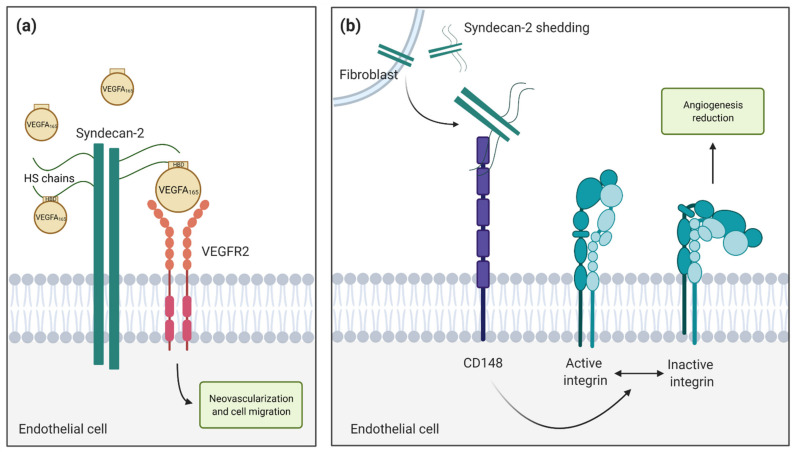
Roles of syndecan-2 in promoting and inhibiting angiogenesis. (**a**) Syndecan-2-VEGFA_165_-VEGFR2 tricomplex promotes angiogenesis. Heparan sulfate chains present in syndecan-2 associate to VEGFA_165_ through a heparin binding domain (HBD) increasing its local concentration and enhancing the binding to its receptor VEGFR2 (**b**) Syndecan-2 shedding inhibits angiogenesis. Shed syndecan-2 binds to CD148 which promotes the reduction of β1-integrin activity in endothelial cells. The inactivation of β1-integrin reduces cell migration and consequently inhibits angiogenesis.

**Table 1 biomolecules-11-00349-t001:** Syndecans location and principal functions as transmembrane receptors.

Syndecan	Main Location	Function
Syndecan-1	Epithelial and plasma cells [16]	Cooperates with several integrins (αvβ3, αvβ5, α2β1, α3β1, and α6β4) through the core protein. Plays roles in matrix remodeling, cell adhesion and spreading, migration, and cytoskeleton rearrangements [17]. Present in breast [18,19], prostate [20], colorectal [21], and pancreatic [22,23] cancers
Syndecan-2	Mesenchymal cells [24]	Important regulator of angiogenesis [25,26]. Present in some cancers such as PDAC [27] and colon cancer [28]. Regulates actin cytoskeleton organization, especially in lung cancer [29]
Syndecan-3	Brain, nervous system, and cartilage [30,31]	Important for brain development as well as in feeding behaviors (is upregulated in the hypothalamus in response to food deprivation] [32]. Plays roles in rheumatoid arthritis disease [33], angiogenesis [34], and HIV-1 infection [35]
Syndecan-4	Most tissues [36,37]	Plays roles in cell mechanics [38] and induces focal adhesion formation [39] and cytoskeleton organization [40,41]

## Data Availability

The data presented in this study are openly available.

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
