# Peer review of "Syndecans and Pancreatic Ductal Adenocarcinoma"

_biomolecules, 2021, doi:10.3390/biom11030349_

Round 1

Reviewer 1 Report

In this review entitled “Syndecans in Pancreatic Ductal Adenocarcinoma”, presented by Nausika Betriu et al., the authors addressed their attention to the multiple physiological aspects and molecular proprieties of Syndecans, a family of transmembrane heparan sulfate proteoglycans, involved in cell adhesion, motility, proliferation, and differentiation processes. Furthermore, they summarized well how Syndecans and their soluble counterparts affect cancer cells' behavior and tumor microenvironment during cancer progression. More importantly, in this review, they described the key roles of Syndecan-1 and Syndecan-2 in pancreatic ductal adenocarcinoma progression.

Overall, this review is well written. However, it deserves a minor revision to address the two issues reported below.

- In the third section of the manuscript (i.e., “Syndecan shedding”), the authors should provide a brief description of how the syndecan soluble molecules, when released from the surface of the cell membrane, can be involved in bacterial pathogenesis.

- The authors should provide a brief overview of how Syndecans, when stocked as cargo within exosomes, may influence the tumor microenvironment's behavior within the tumor niche and distant sites.

Author Response

In this review entitled “Syndecans in Pancreatic Ductal Adenocarcinoma”, presented by Nausika Betriu et al., the authors addressed their attention to the multiple physiological aspects and molecular proprieties of Syndecans, a family of transmembrane heparan sulfate proteoglycans, involved in cell adhesion, motility, proliferation, and differentiation processes. Furthermore, they summarized well how Syndecans and their soluble counterparts affect cancer cells' behavior and tumor microenvironment during cancer progression. More importantly, in this review, they described the key roles of Syndecan-1 and Syndecan-2 in pancreatic ductal adenocarcinoma progression.

Overall, this review is well written. However, it deserves a minor revision to address the two issues reported below.

- In the third section of the manuscript (i.e., “Syndecan shedding”), the authors should provide a brief description of how the syndecan soluble molecules, when released from the surface of the cell membrane, can be involved in bacterial pathogenesis.

As recommended by the reviewer, we have added a brief description of syndecan shedding in bacterial pathogenesis (see section 3.3, lines 311-322)

- The authors should provide a brief overview of how Syndecans, when stocked as cargo within exosomes, may influence the tumor microenvironment's behavior within the tumor niche and distant sites.

Added in Lines 139-144

Reviewer 2 Report

This generally well written manuscript focuses on the many and complex interactions of the 4 syndecans, a protein familiy which is in the focus of many recent publications. This manuscript for ‚Biomolecules‘ has an with an emphasis on protein structure. It includes references to many tumor types and physiological aspects, yet this sometimes leads to hard comprehension. While the aim of the paper ist o give an outlook on PDAC treatment, much of the manuscript is (important) background on syndecans. In addition, all 4 syndecans are discribed, yet the abstract promises a focus on syndecans 1 and 2.

This manuscript on an protein family coming more and more in focus of tumor research and especially diagnosis/prognosis would greatly benefit from additional figures modelling effects of cellular interactions and with an emphasis on localisation (tumor cell, stromal cell, ECM) and form (transmembrane vs shedded). These complex interactions discribed in the text, can lead to very differentiated effects of overexpression vs. low expression of syndecans with. An additional figure regarding angiogensis would be helpfull.

An important aspect for ‚clinical‘ readers (and the visibility of the paper) would be to include CD138  as alternative name for syndecan1. Checking for CD138 targeted cancer therapy, clinical trials with Indatuximab/ravtansine and CAR Tcells come up (multiple myeloma). The authors should comment on this.

This manuscript would benefit from editing the final layout.

Author Response

This generally well written manuscript focuses on the many and complex interactions of the 4 syndecans, a protein familiy which is in the focus of many recent publications. This manuscript for ‚Biomolecules‘ has an with an emphasis on protein structure. It includes references to many tumor types and physiological aspects, yet this sometimes leads to hard comprehension. While the aim of the paper ist o give an outlook on PDAC treatment, much of the manuscript is (important) background on syndecans. In addition, all 4 syndecans are discribed, yet the abstract promises a focus on syndecans 1 and 2.

This manuscript on an protein family coming more and more in focus of tumor research and especially diagnosis/prognosis would greatly benefit from additional figures modelling effects of cellular interactions and with an emphasis on localisation (tumor cell, stromal cell, ECM) and form (transmembrane vs shedded). These complex interactions discribed in the text, can lead to very differentiated effects of overexpression vs. low expression of syndecans with.

An additional figure regarding angiogensis would be helpfull.

As suggested, we have added a new figure in the angiogenesis section (see Figure 5)

An important aspect for ‚clinical‘ readers (and the visibility of the paper) would be to include CD138  as alternative name for syndecan1. Checking for CD138 targeted cancer therapy, clinical trials with Indatuximab/ravtansine and CAR Tcells come up (multiple myeloma). The authors should comment on this.

As recommended, we have included CD138 as an alternative name for syndecan-1 (see line 328). We also have added some information on BT062 clinical trials (see line 338-342 and 591-600)

This manuscript would benefit from editing the final layout.

Reviewer 3 Report

This is an extensive review article on syndecans explaining their structure, their role in signaling, their contribution to disease in particular pancreatic cancer, their implications in angiogenesis and ECM composition. The manuscript is well written. Sometimes on would wish to have less details and not only a summary of numerous papers but also a comprehensive overview and some more critical discussion of the findings.

Other points

1) “Diabetes” should be described more properly as diabetes mellitus

2) Pancreatic dysfunction is NOT a cause of cystic fibrosis. It is vice versa.

3) PDAC desmoplasia should be explained in more detail. Please describe potential causes, the cells responsible for this and in particular the consequences. It would be helpful for many readers to give an overview about how little cancer cells can be found in a highly desmoplastic PDAC stroma.

3) The interaction of syndecans with vegf or vegf receptors are very interesting. This should be described in more detail. Importantly, it needs to be stated which vegf receptor is affected as vegfr1-3 have very distinct functional roles. Also, whenever possible the vegf isoform (A,B,C, A121, A165 …) should be noted. It would also be worthwhile to show this in an additional figure.

4) The phenotypes of syndecan-deficient mice should be presented in an additional table.

5) The authors explain very well the alterations of the ECM. They should pay some attention to how this alters stiffness and to how stiffness can affect tumor progression.

Author Response

This is an extensive review article on syndecans explaining their structure, their role in signaling, their contribution to disease in particular pancreatic cancer, their implications in angiogenesis and ECM composition. The manuscript is well written. Sometimes on would wish to have less details and not only a summary of numerous papers but also a comprehensive overview and some more critical discussion of the findings.

Other points

1) “Diabetes” should be described more properly as diabetes mellitus

 As recommended, “diabetes” has been described as “diabetes mellitus” (see lines 35 and 253)

2) Pancreatic dysfunction is NOT a cause of cystic fibrosis. It is vice versa.

 The disease cystic fibrosis has been eliminated from the list of diseases that cause pancreatic dysfunction (see line 35)

3) PDAC desmoplasia should be explained in more detail. Please describe potential causes, the cells responsible for this and in particular the consequences. It would be helpful for many readers to give an overview about how little cancer cells can be found in a highly desmoplastic PDAC stroma.

We have added some information about the desmoplastic stroma (which cells are the responsible of producing it, chemotherapeutic resistance as a consequence and also that the stroma accounts for the 90% of the tumor mass) (see lines 74-80)

3) The interaction of syndecans with vegf or vegf receptors are very interesting. This should be described in more detail. Importantly, it needs to be stated which vegf receptor is affected as vegfr1-3 have very distinct functional roles. Also, whenever possible the vegf isoform (A,B,C, A121, A165 …) should be noted. It would also be worthwhile to show this in an additional figure.

 In the manuscript, we focus on the interaction between the VEGFR2 and VEGFA165 with syndecan-2 and the isoforms are specified in the text (see lines 514, 517, 533). As suggested, we have added a new figure in the angiogenesis section (see Figure 5)

4) The phenotypes of syndecan-deficient mice should be presented in an additional table.

As recommended, we have added a table summarizing knock out experiments in mice models (see line 538, Table 2)

5) The authors explain very well the alterations of the ECM. They should pay some attention to how this alters stiffness and to how stiffness can affect tumor progression

In the manuscript we have mentioned that PDAC tissue is several folds stiffer than normal pancreatic tissue and we do agree that this is a very important point for cancer disease progression. There is a lot of work done in studying the effect of stiffness mainly in breast cancer, but less is known related to pancreatic cancer. We are actually working on exploring the effect of the stiffness on pancreatic cancer cells phenotype in 3D cell cultures using synthetic self-assembling peptides scaffolds, so this topic is very interesting to us. However, because this manuscript is focused on syndecans and to our knowledge there are no publications describing the effect of matrix stiffness on syndecan biology, we consider that it is better not to include matrix stiffness in this review.

Round 2

Reviewer 2 Report

The authors made some improvements on this generally well written manuscript. Especially including a new figure on angiogenesis and referencing clinical aspects of syndecan1 (alias CD138).

Obviously the manuscript still decribes the many and complex interactions of the syndecan family and references many tumor types. Therefore the promised focus on syndecan 1 and 2 is still not met totally (and doesn’t need to). In addition it seems that the manuscript cannot show ‚Syndecans in PDAC‘ but rather focuses on the proscects of syndecans for PDAC. So this reviewer suggests to adjust the abstract and title accordingly for these two points. (maybe ‚Syndecans and PDAC‘?)

The new reference 54 (a review by Friand et al), contains the word ‚angiogenesis‘ but only references an earlier paper by Zhang and Grizzle (2014).

This manuscript would benefit from editing the final layout. Table 2 seems not optimally placed with regard to the text. Proof reading is required (some typing errors occured while adding text).

Author Response

Dear reviewer,

Thank you for your suggestions. We have made all the necessary changes to be sure we are answering your suggestions properly. Please, see our answer below each of your comments:

Review comment 1:

The authors made some improvements on this generally well written manuscript. Especially including a new figure on angiogenesis and referencing clinical aspects of syndecan1 (alias CD138).

Obviously the manuscript still decribes the many and complex interactions of the syndecan family and references many tumor types. Therefore the promised focus on syndecan 1 and 2 is still not met totally (and doesn’t need to). In addition it seems that the manuscript cannot show ‚Syndecans in PDAC‘ but rather focuses on the proscects of syndecans for PDAC. So this reviewer suggests to adjust the abstract and title accordingly for these two points. (maybe ‚Syndecans and PDAC‘?)

Answer 1: The manuscript title has been changed to: Syndecans and PDAC (see line 2). In addition, the abstract was changed accordingly (lines: 16, 19-22). Please, see the track changes on the right side.

Reviewer comment 2:

The new reference 54 (a review by Friand et al), contains the word ‚angiogenesis‘ but only references an earlier paper by Zhang and Grizzle (2014).

Answer 2: Reference 54 has been changed to Zhang and Grizzle 2014, as suggested (lines 773-774).

Reviewer comment 3:

This manuscript would benefit from editing the final layout. Table 2 seems not optimally placed with regard to the text. Proof reading is required (some typing errors occured while adding text).

Answer 3: The final layout (6. Future perspectives) has been revised carefully and only two changes were needed (line 632, 643). In addition, Table 2 was placed in a more appropriate place into the manuscript (between line 347 and 348).

Finally, please see the new manuscript version attached to this reply to the Reviewer Report (Reviewer 2). 
